# The Roles of Psychological Inflexibility and Mindful Awareness on Distress in a Convenience Sample of Black American Adults in the United States

**DOI:** 10.3390/bs15020112

**Published:** 2025-01-22

**Authors:** Akihiko Masuda, Bradley L. Goodnight, Nicole E. Caporino, Cerila C. Rapadas, Erin C. Tully

**Affiliations:** 1Department of Psychology, University of Hawaiʻi at Mānoa, Honolulu, HI 96822, USA; cerilar@hawaii.edu; 2Department of Psychology, Georgia State University, Atlanta, GA 30302, USA; bradley.l.goodnight@gmail.com (B.L.G.); etully2@gsu.edu (E.C.T.); 3Department of Psychology, American University, Washington, DC 20016, USA; caporino@american.edu

**Keywords:** black, African Americans, psychological inflexibility, mindfulness, experiential avoidance

## Abstract

Background: In recent years, the conceptual framework of psychological flexibility/inflexibility has been of global interest in the field of behavioral health. Nevertheless, studies and evidence of psychological flexibility/inflexibility remain limited for underrepresented groups of individuals, including people of color in the United States (U.S.). Among these groups of individuals are Black Americans in the U.S. In response to this empirical gap, the present cross-sectional study investigated whether psychological inflexibility and mindful awareness were uniquely related to general psychological distress, somatization, depression, and anxiety in Black American adults in the United States. Methods: A convenience sample of 359 Black American college students completed self-report measures of interest online. Results: As predicted, correlational analyses showed that psychological inflexibility was positively associated with general psychological distress, somatization, depression, and anxiety, and that mindful awareness was negatively associated with these four distress variables. A path analysis model revealed that psychological inflexibility, but not mindful awareness, was uniquely associated with these distress variables. Conclusions: The present study extended previous findings with a convenience sample of Black American college students, suggesting that psychological inflexibility may be a useful construct for understanding psychological distress, more so than mindful awareness, among Black American adults in the U.S.

## 1. Introduction

In recent years, the concept of psychological flexibility/inflexibility has been of global interest in the field of behavioral health (e.g., [23]; [56]). Nevertheless, studies and evidence of psychological flexibility/inflexibility remain limited for underrepresented groups, including the people of color in the United States (U.S.; [11]; [23]; [41]). Relevant to the present investigation, Black Americans in the U.S. have been considered one of the underrepresented groups in behavioral health research in general (e.g., [28]) and in psychological flexibility/inflexibility research in particular (e.g., [2]; [23]; [45]).

To date, various psychosocial factors have been linked to psychological distress in Black Americans, including economic hardships, experiences of discrimination, limited social support, and exposure to violence (e.g., [6]; [31]; [36]). Additionally, extant findings show that emotion and behavior regulation, the ways in which an individual responds to internal and external events, are also associated with a range of distress variables in this group ([3]; [15]; [48]). One conceptual framework that emphasizes the role of emotion and behavior regulation on optimal health and wellbeing is the psychological flexibility model ([20], [21], [22]), a contextual behavioral science (CBS) account of behavioral health and intentional behavioral change. Relatedly, in psychological research, psychological inflexibility and mindful awareness are two constructs that are often used to study the aspects of this model with racially diverse samples of adults in the U.S. (e.g., [30]; [40], [39]). Following this conceptual and methodological framework, the present cross-sectional study examined whether psychological inflexibility and mindful awareness were uniquely related to general psychological distress, somatization, depression, and anxiety in a convenience sample of Black American college students in the U.S.

### 1.1. Psychological Inflexibility

Within the CBS model of psychological flexibility, psychological inflexibility is viewed as a transdiagnostic process of psychopathology, characterized by rigid cognitive and behavioral efforts to control and avoid unwanted psychological experiences, combined with behavioral deficits in values-consistent behaviors ([25]; [30]). In research and practice, despite some notable limitations (e.g., [12]; [53]), the Acceptance and Action Questionnaire-II (AAQ-II; [7]) is one of the most widely used self-report measures of psychological flexibility/inflexibility. Today, a large body of evidence shows that, when psychological inflexibility is measured by the AAQ-II, it is consistently and negatively associated with a range of psychological distress variables, including somatization, depression, and anxiety across a wide range of adult samples (e.g., [7]; [43], [44]).

To date, a few cross-sectional studies have investigated psychological flexibility/inflexibility and its link to psychological distress among Black American college students in the U.S. In one of these studies ([37]), psychological flexibility, the direct contrast of psychological inflexibility, was found to be negatively associated with general psychological distress in a convenience sample of 301 Black American college students. This finding was later replicated with another convenience sample of Black American college students (*n* = 561) by using the construct of psychological inflexibility measured by the AAQ-II ([45]).

Furthermore, a more recent study examined the measurement invariance of the AAQ-II across White American, Black American, Latinx American, Asian American, and Middle Eastern American college students in the U.S. ([8]). Relevant to the present investigation, the findings of this study provided evidence for configural invariance (i.e., scores on an instrument load onto the same number and pattern of latent factors) across all racial/ethnic groups, and, as predicted, psychological inflexibility measured by the AAQ-II was positively associated with anxiety and depression across all racial/ethnic groups, including the Black American group (*n* = 1440).

### 1.2. Mindful Awareness

Contrary to psychological inflexibility, mindful awareness has been found to be associated with greater psychological health and wellbeing (e.g., [5]; [9]; [10]). Mindful awareness, which is often measured by the Mindful Attention Awareness Scale (MAAS), refers to the general salutary tendency of being attentive to and aware of the present-moment experience ([9]). Within the psychological flexibility model, this construct is theorized to reflect largely the way an individual recognizes the stream of conscious experience one moment at a time as it is ([25]; [33]). To date, several cross-sectional studies have revealed the conceptual relevance of mindful awareness to Black college students (e.g., [37]; [45]; [46]). In particular, these studies have consistently shown the inverse association of mindful awareness with a range of distress variables (e.g., distress, anxiety, depression, and rumination) in convenience samples of Black American adults in the U.S.

### 1.3. Psychological Inflexibility and Low Mindful Awareness as Unique Pathways to Distress

Following the framework of the psychological flexibility model ([19]; [22], [25]), it is possible that psychological inflexibility and diminished mindful awareness can be understood as two unique pathways to a range of psychological distresses. In fact, pertinent studies with racially diverse college students in the U.S. showed that both psychological inflexibility and mindful awareness were uniquely related to somatization, depression, and anxiety (e.g., [39]; [35]), and these finding were also replicated with a convenience sample of Asian American college students in the U.S. ([40]).

At the time of the present study, no studies had investigated the roles of psychological inflexibility and mindful awareness simultaneously on psychological distress among Black American college students. However, given the above-mentioned findings, it would be reasonable to speculate that these two constructs each would also be uniquely related to a range of distress variables among Black American college students in the U.S. That being said, as discussed extensively elsewhere (e.g., [18]; [26]; [52]), this type of universality assumption should not be made without direct examination with the target sample of interest. As such, the next logical step was to directly investigate the roles of psychological inflexibility and mindful awareness together with a sample of Black American college students in the U.S.

### 1.4. The Present Study

The aim of the present cross-sectional study was to gather preliminary evidence of whether psychological inflexibility and mindful awareness were uniquely associated with general and specific distress variables with a convenience sample of Black college students. Given previous findings (e.g., [8]; [35]), we predicted that psychological inflexibility would be uniquely and positively associated with general distress, somatization, depression, and anxiety. Similarly, based on previous findings ([39]; [35]), we predicted that mindful awareness would be negatively and uniquely associated with these distress variables of interest.

## 2. Methods

### 2.1. Participants

The present cross-sectional investigation was conducted at a public research university in a metropolitan area of the southeastern United States. Potential participants were recruited from undergraduate psychology courses offered at the university through a web-based research tool (i.e., SONA system) from March 2010 to April 2013. These potential participants were 1060 college undergraduates with various racial backgrounds (e.g., *n_woman_* = 816), who voluntarily completed an anonymous web-based survey package, which included the measures of interest. As compensation for their participation, participants received research credits for their undergraduate courses. For the purpose of the present study, data from self-identified Black American students (*n* = 359) were selected for analyses (see Table 1).

Approximately 85% (*n* = 305) of the present participants self-identified as a woman, and 15% identified as a man (*n* = 54). The mean age of the present sample was 20.73 years old (*SD* = 5.17, range 18–57). Approximately 53% of the participants self-identified as being middle class. Roughly 30% of the participants self-identified as being working class, 9% upper-middle class, 8% as poor, and less than 1% as upper class. With respect to sexual identity, one male participant self-identified as being gay, and the remaining male participants self-identified as being straight. Approximately 93% of female participants self-identified as being straight, 5% bisexual, and 2% lesbian.

### 2.2. Measures and Procedures

Approval was obtained from the Institutional Review Board (IRB) of the corresponding author’s institution at the time of the study to ensure adherence to ethical guidelines. The participants were asked to complete an anonymous web-based survey package that included the measures of interest. More specifically, once consented, participants provided demographic information and completed the following measures of mindful awareness, psychological inflexibility, and psychological problems.

#### 2.2.1. Mindful Awareness

The Mindful Attention Awareness Scale (MAAS; [9]) is a 15-item measure designed for participants to self-report the frequency of their attention and alertness to internal and external events in the present moment. Sample items include “It seems I am ‘running on automatic’, without much awareness of what I’m doing” and “I rush through activities without being really attentive to them”. The questionnaire uses a 6-point Likert-type scale (i.e., 1 = *almost always*; 2 = *very frequently*; 3 = *somewhat frequently*; 4 = *somewhat infrequently*; 5 = *very infrequently*; 6 = *almost never*). Higher scores on the questionnaire indicate greater mindful awareness. According to [9] ([9]), Cronbach’s alpha for this measure generally ranged from 0.80 to 0.90. Previous studies showed that Cronbach’s alpha for this measure ranged from 0.88 to 0.90 in a sample of Black college students ([37]; [45]). In the present study, Cronbach’s alpha for the MAAS was 0.90.

#### 2.2.2. Psychological Inflexibility

The Acceptance and Action Questionnaire-II (AAQ-II; [7]) is a 7-item questionnaire designed to assess acceptance, experiential avoidance, and psychological inflexibility. Following a CBS account of psychological flexibility ([30]; [39]), AAQ-II items measure psychological inflexibility in the dimension of activeness (e.g., “My painful experiences and memories make it difficult for me to live a life that I would value”) and that of openness (e.g., “I worry about not being able to control my worries and feelings”). The AAQ-II uses a 7-point Likert-type scale ranging from 1 (*never true*) to 7 (*always true*) to compute the composite score. Higher scores on the questionnaire reflect higher levels of experiential avoidance and inactiveness. Lower scores on the AAQ-II reflect less experiential avoidance and higher levels of acceptance and action. Internal consistency for this measure generally ranged from 0.78 to 0.88. A previous study showed that Cronbach’s alpha for this measure was 0.93 in a sample of Black college students ([45]). In the present study, Cronbach’s alpha of AAQ-II was 0.93.

#### 2.2.3. Depression, Anxiety, and Somatic Symptoms

The Brief Symptom Inventory 18 (BSI-18; [13]) is an 18-item self-report measure designed to assess major forms of psychological distress: depression (e.g., feeling no interest in things, hopeless, or unhappy), anxiety (e.g., feeling uneasy or tense), and somatization (e.g., feeling faintness or dizziness). The self-report measure uses a 5-point Likert-type scale ranging from 0 (*not at all*) to 4 (*extremely*). Higher scores on the BSI-18 subscales reflect higher levels of depression, anxiety, and somatic symptoms. Internal consistency for the depression, anxiety, and somatization subscales in an ethnically diverse sample of college students ranged from 0.78 to 0.87 ([44]). In the present study, Cronbach’s alphas ranged from 0.84 to 0.85.

#### 2.2.4. General Distress

The General Health Questionnaire-12 (GHQ-12; [14]) was used to assess current and nonclinical psychological distress (e.g., feeling useless, incapable, strained). The questionnaire uses a 4-point Likert-type scale ranging from 0 (*not at all*) to 3 (*much more than usual*). Greater scores reflect more severe mental health difficulties. Internal consistency for the GHQ-12 has ranged from 0.87 to 0.89 ([38]; [35]). In samples of Black college students, it has ranged from 0.87 to 0.90 ([37]; [45]). In the present study, Cronbach’s alpha for the scale was 0.86.

### 2.3. Analytic Plan

Pearson correlations were used to evaluate the strength of association between study variables. The hypotheses were tested using path analysis in R statistics version 3.4.3 with RStudio version 1.1.353 and lavaan package version 0.5–23.1097 ([47]). Path analysis, which is an extension of multiple regression analysis that allows for simultaneous testing of multiple associations, was used so that all four distress variables could be included in the same model. The four distress variables were regressed on psychological inflexibility and mindful awareness. Since previous research has shown that female gender and younger age are associated with greater psychological distress in a Black college student sample ([37]), and that college students who identify as a sexual minority have greater psychological inflexibility and/or lower mindful awareness ([42], [39]), participant gender, age, and sexual identity/orientation were included as covariates in the full model. Additional analyses were conducted to determine whether the exclusion of covariates would alter the pattern of results and are presented at the end of the results section.

## 3. Results

There were no missing data. Assumption tests, including tests for multicollinearity and a visual inspection of the normal distribution of residuals, showed that the data patterns met the assumptions of regression.

### 3.1. Descriptive Statistics

Descriptive statistics are presented in Table 2. The bivariate associations between total scores for study variables show that psychological inflexibility was positively associated with general distress, somatization, depression, and anxiety and that mindful awareness was negatively associated with all four distress variables. Psychological inflexibility and mindful awareness were inversely related.

### 3.2. Path Analysis

In the full path model that included all variables, only the covariate of sexual identity/orientation (0 = being straight, 1 = sexual minority) had a significant relationship with distress variables: depression, *B* = 0.37, *SE* = 0.14, *p* = 0.01, and anxiety, *B* = 0.31, *SE* = 0.13, *p* = 0.01, with individuals identifying as a sexual minority reporting higher levels of depression and anxiety. Sexual identity/orientation was not significantly associated with general psychological distress or somatization. Age and gender (0 = man, 1 = woman) were not significantly associated with any of the distress variables in the full model.

As predicted, higher psychological inflexibility was significantly associated with higher levels of all four distress variables. Contrary to our hypothesis, mindful awareness was not significantly associated with any of the distress variables. Figure 1 displays standardized coefficients for the hypothesized effects. Full results, including *R*^2^ effect sizes, are presented in Table 3.

Two additional analyses were conducted to examine the robustness of the model. These analyses include: (1) a path analysis in which the covariates of age, gender, and sexual orientation were excluded and (2) a path analysis that excluded all non-heterosexual participants. Both additional models resulted in patterns of associations that are consistent with the original (full) path analysis. Specifically, estimates of the four distress variables regressed on psychological inflexibility were significant and moderate in magnitude in both additional models, and the four parameter estimates for mindful awareness predicting the distress variables were non-significant and very small in magnitude in both models.

## 4. Discussion

Black American adults in the U.S. continue to be underrepresented samples in psychological flexibility research (e.g., [23]; [32]). In response to this empirical/social justice gap, the present cross-sectional study investigated whether psychological inflexibility and mindful awareness were uniquely associated with general distress, somatization, depression, and anxiety in a convenience sample of Black American college students in the U.S. As predicted, psychological inflexibility was uniquely and positively associated with all four distress variables when controlling for age, sexual identity/orientation, and mindful awareness. However, contrary to our prediction, mindful awareness was not found to be a unique correlate of general distress, somatization, depression, or anxiety when controlling for key demographic variables and psychological inflexibility.

Our findings partially supported the applicability of the CBS model of psychological flexibility to the present Black American college sample. More specifically, consistent with the model ([25]; [34]), our findings suggest that general distress, somatization, depression, and anxiety in the present Black American sample could be understood using the conceptual framework of psychological inflexibility. These findings also concur with a previous cross-sectional study with Black American college students, which showed a significant positive association between psychological inflexibility and distress variables ([45]). However, as noted above, we found that mindful awareness was not significantly associated with any distress outcomes in the present Black American sample when controlling for psychological inflexibility. This latter finding was somewhat surprising, as prior theories have posited that mindful awareness is fundamental to greater behavioral health and wellbeing ([4], [5]; [10]; [39]). Furthermore, previous studies with Black college samples demonstrated an inverse relationship of mindful awareness and general psychological distress (e.g., [37]; [45]).

Together, our findings may imply that the construct of psychological inflexibility captured by the score of AAQ-II is particularly relevant to a range of distress experienced by Black college students more so than mindful awareness as measured by the MAAS. Clinically, if this is the case, it is possible to speculate that, at least within the present convenience sample of Black American college students, the dimensions/aspects of psychological inflexibility that do not overlap with the construct of mindful awareness account for the unique associations between psychological inflexibility and the distress variables. Relatedly, previous studies suggest that psychological inflexibility measured by the AAQ-II reflects the dimensions of openness (acceptance) and behavioral engagement, which are not necessarily captured by the construct of mindful awareness ([30]; [39]). Thus, these openness and engagement dimensions in psychological inflexibility (e.g., greater avoidance, entanglement, lack of values-directed living) may explain its unique association with distress variables in the present Black American sample that were not found in mindful awareness. Alternatively, our findings may also suggest that the significant association exists between mindful awareness and distress variables, but that this association unfolds in part through psychological inflexibility. If this is the case, mindful awareness and psychological distress should be understood in its association with or through the whole CBS framework of behavioral health and psychological flexibility (i.e., awareness, openness, and engagement; [25]).

### Future Studies

The present findings point to several directions for future research. First, future replication studies should use recently developed and more psychometrically sound measures of psychological inflexibility and mindful awareness. As briefly discussed above, the AAQ-II ([7]) and MAAS ([9]), the self-report instruments that we used to measure psychological inflexibility and mindful awareness, respectively, have been criticized in recent years by some for not fully tapping the constructs they intend to measure (e.g., [12]; [55]). As such, future studies should examine whether they replicate our findings using more psychometrically sound measures of psychological flexibility and/or mindfulness, such as the subscales of Multidimensional Psychological Flexibility Inventory (MPFI; [50]) and Five Facet Mindfulness Questionnaire (FFMQ; [5]). Relatedly, future studies should examine whether our findings can be replicated with other samples of Black American adults ([11]; [23]). In sum, if these future studies provide further support that mindful awareness itself is not significantly related to distress outcomes when psychological inflexibility is considered, it will be important to investigate the factors that might explain the differential roles of psychological inflexibility and mindful awareness.

Second, it is imperative to test the CBS model of behavioral health and psychological flexibility within the context of other known risk and resilient factors, such as economic hardships, perceived discrimination and racism, and social support. This future direction is to examine whether constructs, such as psychological inflexibility and mindful awareness, offer incremental account for better understanding the psychological distress of Black American adults ([27]). Particularly relevant to the present study, one cross-sectional study found that psychological inflexibility and mindful awareness were unique correlate of distress variables when perceived discrimination and ethnic identity were taken into consideration in a racially diverse sample of U.S. college students ([39]), suggesting an incremental value of CBS-related constructs. Relatedly, one cross-sectional study found that mindfulness, a construct relevant to psychological flexibility and inflexibility, moderated the relationship between experience of racism and anxiety in a convenience sample of Black American undergraduate and graduate students ([15]). This latter finding was then replicated with a community sample of racially and ethnically diverse adults ([51]).

Third and relatedly, future studies should also explore the intersection between psychological flexibility, mindful awareness, and coping styles and resources in Black college students ([16]). The constructs of psychological flexibility and mindful awareness, measured by the AAQ-II and MAAS, respectively, seem to parallel some of the effective Afrocultural coping mechanisms, such as cognitive and emotional debriefing and spiritual-centered coping, which are practiced in many Black college students in the U.S. ([16]; [54]). As such, investigating whether adaptive emotion/behavior regulation strategies in the CBS model are redundant with other established Afrocultural frameworks is imperative to the understanding and promotion of behavioral health of Black American adults in the U.S.

Finally, from a perspective of culturally responsive approach to behavioral health ([26]), it is important to investigate how the processes of psychological inflexibility and mindful awareness manifest topographically in Black American adults. This is because the literature suggests that individuals from racial and ethnic minority backgrounds engage in coping strategies that diverge in form from strategies endorsed by White American samples for achieving the same ends ([29]) while these groups are also known to use similar coping strategies to achieve different ends ([16]). The present study showed that the process of psychological inflexibility is relevant to the psychological distress of the present Black American sample, but it does not show how it is manifested topographically.

## 5. Limitations

The current study has several notable limitations. First, due to the convenience sampling of Black college students recruited from a single university in the southeastern United States, our results may not generalize to the general U.S. Black Americans across all age groups. Second, the present study grouped all Black students together into one category. As “Black” represents people whose ancestors originated in a variety of regions with vastly different cultural values and practices (e.g., Caribbean Black individuals and African Americans), creating one label might have obscured important variability within the group ([1]). Third, because of its preliminary nature, the variables included in this study were purposefully limited. As noted above, future research should incorporate factors that are found to be relevant to the behavioral health of Black Americans in the U.S., such as experiences of discrimination, acculturation stress, racism, and social support (e.g., [6]; [31]; [36]), into the models to be tested. Fourth, the AAQ-II and the MAAS, the measures used in the present study have not been fully validated in Black samples, although preliminary findings are available ([45]). Relatedly, the AAQ-II and MAAS have been criticized for their limited validity (e.g., [17]; [49]; [53]; [55]; [57]). In particular, the poor discriminant validity of the AAQ-II vis à vis neuroticism/general psychological distress is concerning as statistically significant relations found between AAQ-II and BSI/GHQ variables in the present study may be artificially inflated due to problematic multicollinearity. As such, the present findings should be interpreted with caution and future studies should replicate our finding with more psychometrically sound measures. Lastly, our guiding CBS model postulates that the regulation process of openness, centeredness, and engagement are dynamic and contextual in nature ([25], [24]). However, causal and directional links could not be drawn because this study is cross-sectional in nature. As such, future studies should investigate whether there are causal, bidirectional, and prospective links between mindful awareness, psychological inflexibility, and distress variables.

## 6. Conclusions

Despite these limitations, the present study extends the literature by partially supporting the CBS model of behavioral health to Black college students in the U.S., further emphasizing the importance of investigating psychological constructs and theories in underrepresented populations. Specifically, we also suggest further investigation of the CBS model, especially the role of psychological inflexibility and mindful awareness, in this population.

## Figures and Tables

**Figure 1 behavsci-15-00112-f001:**
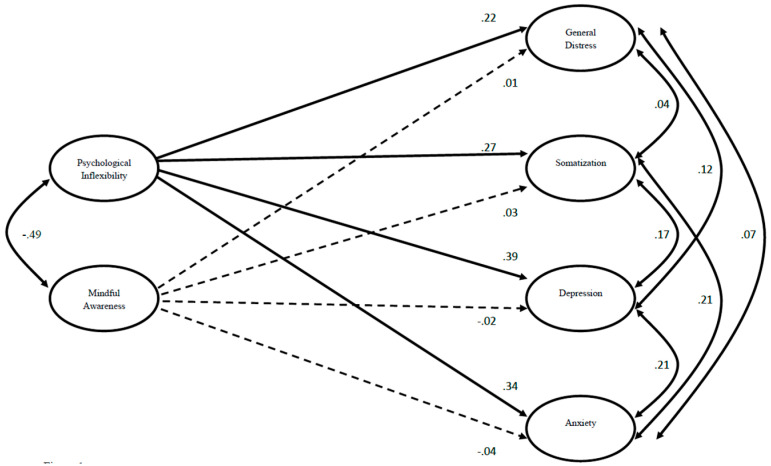
Path Analysis Model for Four Distress Variables Regressed on Psychological Inflexibility and Acting with Awareness. Note: Coefficients are standardized (betas); dashed lines indicate non-significant effects; coefficients control for the effects of age, gender (male = 0; female = 1), and sexual orientation (0 heterosexual = 0; 1 = sexual minority) are not included in the figure but were included in the model and are reported in the text. *n* = 359.

**Table 1 behavsci-15-00112-t001:** Demographic background.

Characteristic	Total (*n* = 359)	
	*M*	*SD*
**Age**	20.73	5.17
	**Percent**	** *n* **
**Gender**		
Man	15.1	54
Woman	84.9	305
**Sexual Identity/Orientation**		
Straight	94.2	338
Gay/Lesbian	1.9	7
Bisexual	3.9	14
**Family Background**		
Poor	7.8	28
Working class	29.8	107
Middle class	52.9	190
Upper-middle class	9.2	33
Wealthy	0.3	1

**Table 2 behavsci-15-00112-t002:** Means, standard deviations, coefficient alphas, and zero-order relations between all variables (*n* = 359).

Variables	1	2	3	4	5	6	7
1. General Distress (GHQ)	--						
2. Somatization (BSI-18 Somatization)	0.44 **	--					
3. Depression (BSI-18 Depression)	0.67 **	0.65 **	--				
4. Anxiety (BSI-18 Anxiety)	0.57 **	0.75 **	0.78 **	--			
5. Psychological Inflexibility (AAQ-II)	0.59 **	0.55 **	0.66 **	0.64 **	--		
6. Acting with Awareness (MAAS)	−0.22 **	−0.28 **	−0.29 **	−0.30 **	−0.41 **	--	
7. Age	−0.09	−0.13 *	−0.13 *	−0.10	−0.13 *	0.13 *	--
*M*	11.53	3.38	4.59	3.72	19.08	59.16	20.73
*SD*	6.19	4.29	4.94	4.49	9.47	13.64	5.17

Note: * *p* < 0.05, ** *p* < 0.01, GHQ = General Distress Questionnaire; BSI-18 = Brief Symptom Inventory-18 item; AAQ-II = Acceptance and Action Questionnaire-II; MAAS = Mindfulness Attention Awareness Scale; correlations are between sum-scores computed using all items; relationships with binary control variables sexual orientation and gender are reported in the text.

**Table 3 behavsci-15-00112-t003:** Path analysis results for full model with all variables and covariates included.

Covariates Only	Outcome															
	General Distress				Depression				Somatization				Anxiety			
	*B*	*SE*	β	*p*	*B*	*SE*	β	*p*	*B*	*SE*	β	*p*	*B*	*SE*	β	*p*
Age	−0.01	0.01	−0.09	0.08	**−0.02**	**0.01**	**−0.14**	**0.01**	**−0.02**	**0.01**	**−0.13**	**0.01**	**−0.02**	**0.01**	**−0.10**	**0.05**
Gender	0.07	0.08	0.05	0.39	0.11	0.11	0.06	0.29	0.13	0.12	0.06	0.29	0.11	0.11	0.05	0.31
Sexual Identity/Orientation	0.21	0.12	0.10	0.07	0.21	0.16	0.07	0.20	**0.46**	**0.18**	**0.13**	**0.01**	**0.40**	**0.17**	**0.13**	**0.02**
*R* ^2^	0.74				0.50				0.35				0.46			
All Variables																
Age	0.00	0.00	−0.02	0.73	−0.01	0.01	−0.04	0.32	−0.01	0.01	−0.06	0.19	0.00	0.01	−0.01	0.73
Gender	0.01	0.06	0.00	0.93	0.03	0.09	0.01	0.73	0.05	0.09	0.03	0.56	0.03	0.09	0.02	0.71
Sexual Identity/Orientation	0.16	0.09	0.07	0.09	**0.37**	**0.14**	**0.11**	**0.01**	0.14	0.13	0.05	0.31	**0.31**	**0.13**	**0.10**	**0.01**
Psychological Inflexibility	**0.22**	**0.02**	**0.59**	**0.00**	**0.39**	**0.03**	**0.64**	**0.00**	**0.27**	**0.03**	**0.51**	**0.00**	**0.34**	**0.02**	**0.62**	**0.00**
Mindful Awareness	0.01	0.03	0.02	0.63	−0.02	0.04	−0.02	0.62	−0.05	0.04	−0.07	0.16	−0.04	0.04	−0.05	0.24
*R* ^2^	0.83				0.50				0.35				0.46			
Δ*R*^2^	0.09				0.15				0.28				0.22			

Note. *n* = 359; Significant associations highlighted in bold (*p* < 0.05); Δ*R*^2^ is the percent variance in outcome accounted for after accounting for covariates; Gender (0 = male, 1 = female). Sexual orientation (0 = heterosexual, 1 = sexual minority); bold values = significant findings.

## Data Availability

The datasets generated during and/or analyzed during the current study are available from the corresponding author on reasonable request.

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
