# Peer review of "The Roles of Psychological Inflexibility and Mindful Awareness on Distress in a Convenience Sample of Black American Adults in the United States"

_behavsci, 2025, doi:10.3390/bs15020112_

Round 1
Reviewer 1 Report
Comments and Suggestions for Authors
Thank you for the opportunity to review the manuscript ‘The roles of psychological inflexibility and mindful awareness in distress in a convenience sample of Black American adults in the US.’ The paper is generally very well-written, the analyses robust, and the topic one of relevance to fostering well-being and health in underrepresented groups. I believe that with the incorporation of the minor suggestions I offer below, this paper will make a valuable contribution to the literature.
Introduction:
-The introduction is generally very well-written and makes a strong argument for the relevance of the proposed analyses. The conceptual foundations of variables of interest (psychological inflexibility and mindful awareness) are described accurately, and relevant literature in this area provides a comprehensive review of past work germane to the present study. I do not say this very often, but I do not have any recommended edits for the intro- nice work!
Method:
-What was the date range of data collection for the study? Would be helpful to add this if possible to provide greater context for the sample.
-For the BSI-18 sub-section, the authors should describe the valence of scores, similar to what is done for the MAAS and AAQ-II (e.g., “higher MAAS scores indicate greater mindful awareness”). This will help the reader interpret findings of directional statistical associations.
Results:
-Findings are presented in a logical, clear manner and can be easily interpreted.
Discussion:
-Page 8/13, line 311: sentence beginning with “replication studies with recently developed and more psychometrically sound measures...” is incomplete and lacking a concluding clause.
-In the final sentences of the ‘limitations’ paragraph the authors nicely describe the dynamic and contextual nature of psychological flexibility-related variables. I also think it would be worth being a bit more explicit about how causal/directional inferences cannot be made using a cross-sectional design.
-The discussion section is also well-written and interprets the findings in the context of existing literature, providing interesting speculations for why psychological inflexibility (but not mindful awareness) was related to distress variables. One potential interpretation of the findings that is missing (and I believe worth considering) is in regard to the poor discriminant validity of the AAQ-II vis a vis neuroticism/general psychological distress. The authors nicely highlight the psychometric limitations of the AAQ-II throughout the paper, but I wonder if going a bit more in depth about how the aforementioned discriminant validity issues could have influenced the relations between AAQ-II and distress variables would be warranted? That is, many authors (Wolgast, Tyndall, Rochefort, Arch et al., 2022 special issue of BT) who have critically examined the AAQ-II seem to colloquially describe it as more of a measure of neuroticism/general distress than a measure of PF, per se. Following this reasoning, statistically significant relations between AAQ-II and BSI/GHQ distress variables in the present study may be artificially inflated due to this problematic multicollinearity (i.e., conceptual overlap between PF and distress measures).
Author Response
Thank you for reviewing our manuscript very thoroughly. Attached is our response to each of your comments/suggestions.

Reviewer 2 Report
Comments and Suggestions for Authors
The authors use cross-sectional data from 359 Black American college students to look at whether psychological inflexibility and mindful awareness hold similar relationships with measures of psychological distress as has been found in other populations. Psychological flexibility, but not mindful awareness demonstrated the expected relationships with the distress related variables in a Path model. The limitations of the study are well described as well as the implications for future research.
The following comments aim to address key issues in the abstract and improve the impact of this manuscript.
1) Lines19-21: psychological inflexibility (AAQ-II) was positively correlated with the measures of distress consistent with predictions (lines 123-124). Please correct.
2) Lines 21-22: mindful awareness was negatively associated with the four distress variables consistent with predictions (lines 125-126). Please correct.
3) “generalized process of psychopathology” is an unusual turn of phrase, particularly since CBS does not embrace a pathology-based model of psychological distress. It is more common for psychological inflexibility to be referred to as a transdiagnostic process. Consider revision.
4) Table 1: Percent and N should be in bold.
5) Section 2.2.2.: The AAQ-II items are presented as reflecting active/inactive, openness, and acceptance dimensions. This interpretation is not from Bond et al. (2011). Latzman & Masuda (2013) and Masuda et al. (2022) should be cited as the sources for this interpretation.
6) The manuscript will benefit from additional proofreading, particularly at lines: 45, 103, 298, and 366.
Author Response
Thank you so much for thoughtfully reviewing our manuscript. Attached is our response to each of your comments/suggestions.
